# Connection: Digitally Representing Australian Aboriginal Art through the Immersive Virtual Museum Exhibition

**Rui Zhang *** and **Fanke Peng**

UniSA Creative, University of South Australia, Adelaide, SA 5000, Australia; fanke.peng@unisa.edu.au
* Correspondence: rui.zhang@mymail.unisa.edu.au; Tel.: +61-041-401-0032

**Abstract:** In 2022, the National Museum of Australia launched an immersive virtual exhibition of Australian Aboriginal art: *Connection*: *Songlines from Australia's First Peoples*, which was created and produced by Grande Experiences, the same team that produced the multisensory experience *Van Gogh Alive*. The exhibition employs large-scale projections and cutting-edge light and sound technology to offer a mesmerizing glimpse into the intricate network of Australian Aboriginal art, which is an ancient pathway of knowledge that traverses the continent. Serving as the gateway to the Songlines universe, the exhibition invites visitors to delve into the profound spiritual connections with the earth, water, and sky, immersing them in a compellingly rich and thoroughly captivating narrative with a vivid symphony of sound, light, and color. This article examines *Connection* as a digital storytelling platform by exploring the Grande Experiences company's approach to the digital replication of Australian Aboriginal art, with a focus on the connection between humans and nature in immersive exhibition spaces.

**Keywords:** Australian Aboriginal art; digital representation; immersive virtual exhibition; museum

## 1. Introduction

Since the first half of the twentieth century, museums in Australia have been collecting Australian Aboriginal art. In 1958, the New South Wales Art Gallery in Sydney exhibited a collection of funerary logs carved from Arnhem wood and painted with clay pigments by artists from the tropical north. This marked the first inclusion of Aboriginal artworks in an Australian art museum's collection (Goldstein 2013). In recent years, museums have evolved from being mere repositories of esteemed collections to becoming spaces for dialogue and multisensory experiences (Levent and Pascual-Leone 2014). Immersive technologies within museums allow visitors to actively engage in their search for meaning by interacting with exhibitions. Following global trends, Australian museums are at the forefront of digitally representing collections, introducing new technologies, developing innovative learning strategies, and engaging visitors.

This study focuses on a recent immersive exhibition called *Connection: Songlines from Australia's First Peoples*. The exhibition, which took place at the National Museum of Australia in Canberra from 8 June to 9 October 2022, offered a breathtaking immersive experience. It showcased dynamic imagery of Australia's First Peoples, encompassing individuals from remote, rural, and urban areas who engage in both traditional and contemporary art forms throughout the country. The exhibition emphasized their significant contributions to the arts and cultural heritage of Australia. This study investigates how immersive technologies were utilized to establish a museum space that fosters visitor engagement, inclusivity, and interaction. By examining digital representations of Australian Aboriginal art, including its materials, techniques, manufacturing processes, and the designer's narrative, this research demonstrates that immersive virtual experiences deepen the audience's engagement with and comprehension of Australian Aboriginal art. The insights gained from this encourage further development of immersive virtual exhibitions

to provide audiences with rich digital experiences and increased multisensory awareness of Australian Aboriginal art.

## 2. Understanding Australian Aboriginal Art

### 2.1. The Importance of Australian Aboriginal Art

"'Aboriginal' is a broad term that refers to Nations and Traditional Owners of mainland Australia and most of the islands, including Tasmania, Fraser Island, Palm Island, Mornington Island, Groote Eylandt, and Bathurst and Melville Islands" (University of South Australia 2021). In spite of colonization, Aboriginal citizens have been recognized as a distinct group with unique traditions and values with flourishing Aboriginal cultural forms in the arts and popular culture, particularly through the medium of Aboriginal art (Fisher 2016).

Fisher (2016) recalled the precolonial foundations of Aboriginal expressive forms and explored the way modern forms of Aboriginal art have been shaped by the history of colonialism in Australia in the 1980s. European exhibitions of Aboriginal art were first used to justify the invasion of Australia by suggesting that Aboriginal peoples were primitive and threatening to European culture. However, Aboriginal Australians responded by creating art for public sale to support their communities and preserve their heritage (Sutton 1997). In the twentieth century, what was considered 'Aboriginal art' was art made by Aboriginal Australians living in remote regions, because those living 'beyond the frontier' were distinguished by those who were 'contaminated' by modernity. Indeed, Smith (2006) argues that Australian Aboriginal art has undergone a complex process of modernization. Australian Aboriginal art has always been created by Aboriginal people, but its modernization took place as a result of contact with people of European decent. This process may have begun with the transformation of interrelated artefacts from social and religious rituals into marketable commodities. Accordingly, there are two main phases of Australian Aboriginal art—traditional Aboriginal art that was created before colonization, and contemporary Aboriginal art created after colonization. However, these lines are blurry, as many Aboriginal artists still work with traditional methods while remaining influenced by the colonized world. Thus, 'Aboriginal art' today includes paintings on canvas and linen with acrylics and natural ochres; paintings on bark; local art forms such as pearl shell and emu egg carvings, possum-skin cloaks, hollow log coffins, and spirit figures carved or woven; as well as fine crafts such as shell necklaces, ceramics, and fiber art (Fisher 2012). As an Indigenous curator Hetti Perkins suggests, "The possibilities of Aboriginal art practice are infinite and can have relevance and resonance outside their immediate cultural context while maintaining the integrity of speaking from within that context" (Fink and Perkins 1997).

Aboriginal Australians are the proud keepers of arguably the oldest continuous culture on the planet. Their heritage stretches across many different communities, each with its own unique blend of cultures, customs, and languages. Before the European invasion in 1788, there were more than 250 Indigenous nations, each with several clans (Malaspinas et al. 2016). Throughout the history of Australian cultures, art has been an integral part of knowledge systems that include mapping the landscape and making reference to understanding the landscape through the stories and teachings of Aboriginal ancestors (Neale and Kelly 2020). The development of Australian Aboriginal art form has been extensively documented (Myers 2002). Ever since Australian Aboriginal art has been produced for so-called western art markets, art curators (and cultural theorists) have been challenged to accommodate exhibits within (or beyond) the mainstream categories of art and/or culture (Wildburger 2013). Art and culture are meaningful practices that reflect social values and are also able to establish, affirm, or challenge these values (Schirato 2004). At the core of artistic production is the way Australian Aboriginal artists internalize locality, nationality, and indigeneity. Aboriginal art continues to evolve and negotiate with contemporary art around the globe. In order to make Aboriginal art understood by those who are not born

into it, Aboriginal artists have worked intellectually, culturally, artistically, and politically to ensure that it can be seen, experienced, and understood by others (Skerritt et al. 2016).

Australian Aboriginal art has been collected by museums in Australia since the first half of the twentieth century. In 1958, the New South Wales Art Gallery in Sydney displayed a collection of funerary logs carved in Arnhem wood and with pigments made of clay by painters in the tropical north. It was the first time that an Australian art museum had included Aboriginal artworks in its collection (Goldstein 2013). In recent years, museums have moved from repositories of venerable collections to spaces for speech and multisensory experiences (Levent and Pascual-Leone 2014). Immersive technologies in museums enable visitors to actively participate in their search for meaning by actively interacting with exhibitions. Following the global trends, museums in Australia are the mainstream in terms of digital representing collections, introducing new technologies, developing new learning strategies, and engaging visitors, etc.

### 2.2. Opportunities and Challenges in Creating the Immersive Exhibition of Australian Aboriginal Art

The immersive exhibition of Australian Aboriginal art at the National Museum of Australia presented a complex interplay of opportunities and challenges.

### 2.2.1. Opportunities towards Cultural Attraction, Education, and Collaboration via the Immersive Exhibition

Australian Aboriginal art, intricately woven with identity and spirituality, offers a potent mode of expression within the immersive exhibition. *Connection: Songlines from Australia's First Peoples* served as a channel for communicating the deep bond between Aboriginal art and culture, nurturing comprehension and reverence for cultural heritage. It played a crucial role in conserving and commemorating the abundant cultural legacy of Australian Aboriginal peoples. The collaborative and innovative strategies employed contribute to the transformation of traditional Aboriginal art practices, enhancing the cultural panorama. Additionally, it acted as a repository, preserving traditional knowledge, stories, and artistic traditions for the benefit of future generations.

From a practical perspective regarding the museum's public services, the immersive exhibition acted as a notable cultural attraction. Functioning as a dynamic and captivating educational platform, it imparted knowledge about the profound depth and diversity of Australian Aboriginal art and culture via a focus on storytelling. The exhibition played a significant role in fostering intercultural understanding by providing visitors from diverse backgrounds the opportunity to immerse themselves in the stories and perspectives of Aboriginal communities. The incorporation of cutting-edge technologies amplified the exhibition's storytelling and educational capabilities, offering visitors a more dynamic and interactive experience, thereby enhancing their engagement with the exhibition's content.

The collaborative essence of the exhibition engaged Australian Aboriginal artists, cultural experts, and communities in decision-making processes. This collaboration guaranteed a culturally sensitive and accurate representation while offering Aboriginal artists opportunities to explore new technologies and forms of expression. Simultaneously, it attracted the public to the National Museum of Australia, contributing to tourism and the local economy and creating memorable and impactful visits for patrons.

### 2.2.2. Challenges in Balancing Technology and Authenticity

Balancing accessibility and authenticity posed a challenge, particularly with high-tech immersive elements. While engaging for many, these elements might alienate visitors who prefer a more traditional museum experience. The risk of immersive technology overshadowing the art and culture it represents is evident. Overreliance on technology might detract from the authenticity of the experience, underscoring the necessity for a balanced approach.

Effectively addressing these challenges was vital to prevent misinterpretations or superficial perceptions of Aboriginal culture. Ensuring accurate, respectful, and com-

prehensive educational content represented another critical challenge. Navigating the complexities of Aboriginal culture was necessary to offer a meaningful and authentic experience for visitors. Therefore, strategic measures were needed to ensure that the educational content fosters a nuanced understanding.

The following section delves into the exhibition to address these opportunities and challenges from a design perspective.

### 3. The Immersive Experience within *Connection*

#### 3.1. The Digital Representation of Australian Aboriginal Painting Materials

Upon entering the immersive space, visitors are encouraged to freely explore the exhibition at their own pace, engaging with the artworks by sitting, standing, walking, and investigating. The exhibition showcases a multitude of paintings by 80 First Nations artists residing in urban centers, rural areas, and remote communities. These artworks are digitally represented through large-scale projections on walls and floors, animating the paintings in a captivating manner (see Figure 1). The projections and enrichment of the content enhance every element of the artwork, including colors, images, and brushstrokes. Utilizing cutting-edge visual and audio technologies, a range of digital storytelling scenarios unfold in chronological order, providing visitors with a visual experience. For instance, visitors can sit, stand, or walk on ground-based projections, immersing themselves in the changing scenes illuminated by dynamic digital light and shadow. This allows them to observe the materials used in Aboriginal painting and gain a deeper understanding of the enigmatic history and culture of Aboriginal art. Additionally, contextual information such as stories about the creative process and the artists' way of life is presented through background media, including text and audio interpretations. In this informative manner, visitors are invited to partake in an immersive experience, delving into the lives of Aboriginal artists and the stories behind their artistic creations, which are both simple and sublime.

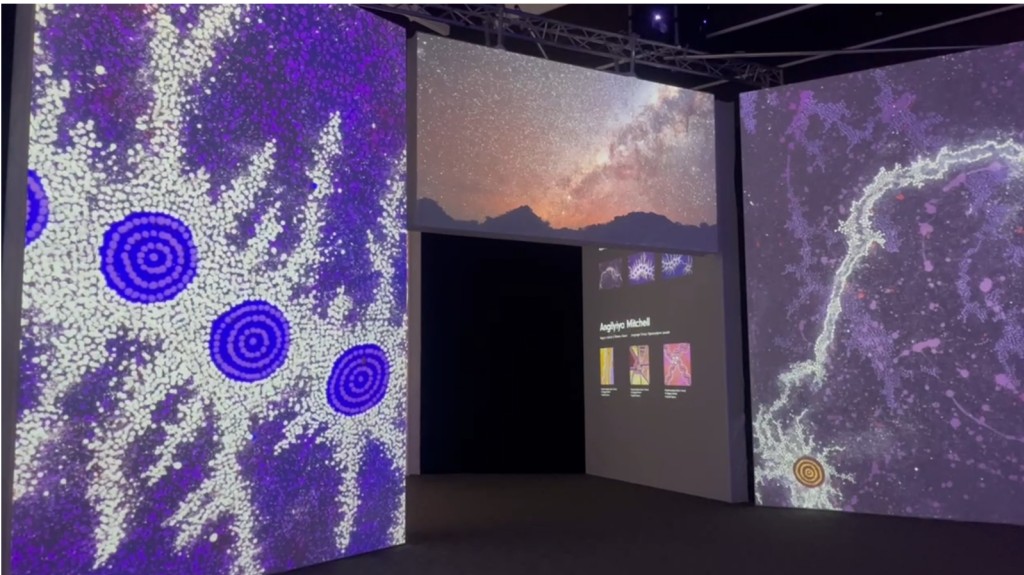

**Figure 1.** The entrance of *Connection: Songlines from Australia's First Peoples* features several large-scale projections that showcase Aboriginal art elements in an ever-changing interplay of light and shadow. Photograph by Rui Zhang, 2022.

Traditionally, Australian Aboriginal artists utilized materials sourced from the natural environment. A diverse range of materials, such as rocks, bark, wood, plant fibers, animal fur, and even human skin, were employed in painting and craft-making. Similar to many other art forms, Australian Aboriginal art involves a deep connection to the origins, usage, and preservation of these materials. Indigenous artists adhere to their own rules and techniques when it comes to handling artistic materials. Understanding these rules and

methods is essential for effectively incorporating immersive technologies into museum exhibitions, ensuring the appropriate application and integration of these technologies.

The utilization of local and distinctive materials in Aboriginal painting relies on the skills, techniques, and expertise of the artists. Immersive technology facilitates the audience's unrestricted movement within an immersive environment, allowing them to experience the material aspects of Aboriginal painting. Through a diverse array of high-definition images, the exhibition showcases the captivating blend of colors and the amalgamation of various materials, highlighting the ingenuity, skills, and expertise of Aboriginal artists. Within an immersive exhibition space, technology serves as a powerful medium to bridge the gap between visitors' experiences and Aboriginal artworks, as the connection between visitors and such artworks may not be immediate. While an experience with the physical artwork may offer a more powerful engagement with materiality, the argument underlying our exploration is that storytelling, particularly in the context of Australian Aboriginal artists' creations, is better communicated through immersive digital experiences. The digital representation of artworks in diverse immersive forms provides visitors with the freedom to choose how to direct their attention and engage their memory, in accordance with their individual cognitive capabilities.

### 3.2. The Digital Storytelling of Australian Aboriginal Artists' Creation

Storytelling serves as a method of interpretation, aiming to inspire internal dialogue and establish a genuine connection with visitors. It also acts as a strategy to create an environment that encourages visitors to construct their own meanings (Bedford 2001). In an interview, Dr. Valerie Keenan, manager of the Girringun Aboriginal Art Centre in northern Australia, highlighted that storytelling has traditionally relied on physical and oral processes such as dance, song, and painting. However, the introduction of new materials and artistic practices represents an extension of the past, as Aboriginal artists explore mediums like ceramics, photography, and video. These explorations enable the continuation of the storytelling tradition while embracing a broader global audience (Butler 2019).

Artworks are typically the result of deliberate human actions, and the production and recognition of art necessitates a diverse range of cognitive skills, including semantic memory and visual perception (De Smedt and De Cruz 2011). Art emerges from normal human perceptual and motivational processes (Barry 2006), and it can be studied and appreciated using cognitive psychology methods as an inherent component of human cognition (Dissanayake 2015). Since the 1980s, Aboriginal art has progressively found its place in the collections of numerous Australian galleries and museums, gaining widespread recognition as a significant aspect of Australian art (McLean 2011). The immersive scenarios presented in *Connection*, adopting a narrative approach, play a vital role in making complex relationships accessible and understandable to visitors (see Figure 2). They also help preserve a sense of identity, particularly in relation to the enigmatic history of Australian Aboriginal culture. Bruner (1993) discussed two characteristics of storytelling that are directly relevant to museum exhibits. Firstly, stories serve as a means of knowledge acquisition, ensuring that human histories and cultures are remembered through narrative construction. Secondly, stories assist in clarifying our core values and beliefs. In the case of *Connection*, the immersive narrative reflects the significance of artistic faith within First Nations communities. By employing multiple scenarios, connections can be forged between the works of First Nations artists and the memories and knowledge of visitors. Rather than being instructed on how to feel about the artworks, visitors are offered a far more powerful experience through the gift of digital storytelling, allowing them to interpret Aboriginal artworks based on their own cognitive processes. In essence, museum curators present a new story of the First Nations people's art, guiding visitors along a path they can design themselves, starting from any digital moment, thus fostering an immersive experience that establishes connections between diverse artistic scenarios through personal associations.

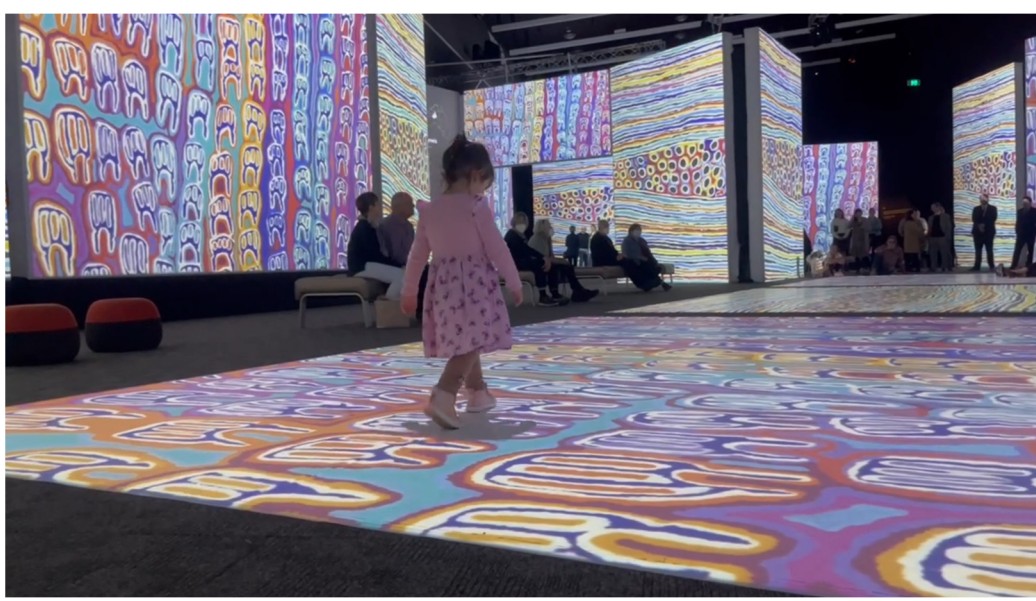

**Figure 2.** A child walks across the expansive floor projection, observing the shifting patterns of light and shadow that showcase various elements of Aboriginal art. Meanwhile, other visitors are either seated or standing, engaged in the exploration of virtual Aboriginal art. Photograph by Rui Zhang, 2022.

*3.3. Visitors Immersive Connection with the Australian Aboriginal Artists' Spirits*

When observing how visitors engage with digitally represented artworks, it becomes evident that their perceptions and understandings of Aboriginal art are shaped by the appropriation and contextualization of immersive technology within Aboriginal cultures, rather than solely by the technology itself. To gain a full appreciation of these immersive art-themed exhibits, it is crucial to examine how they are experienced—how visitors see, interact with, and interpret these works (Mondloch 2022). In comparison to traditional forms of art displayed in museums, *Connection* offers visitors greater freedom and the opportunity to explore Australian Aboriginal art from multiple perspectives. As visitors navigate through the exhibition, they encounter virtual Aboriginal artworks enhanced by narrated voice-overs contributed by the esteemed *Connection* curatorial panel, which includes Elders and leaders in the art world. The *Connection* experience also incorporates a curated soundtrack featuring the works of over 25 First Nations musicians, serving as a pivotal component of the overall experience. However, one issue worth noting and discussing is that at times, the narrations may not be clearly audible due to blending with the soundtrack, despite corresponding texts being displayed on the projections. An approach worth mentioning is found in *Meet Vincent van Gogh*, an immersive traveling exhibition developed by the Van Gogh Museum in the Netherlands. In addition to utilizing common immersive technologies, *Meet Vincent van Gogh* provides visitors with separate versions of the audio guide for adults and children, recognizing the differences in their ability to receive and comprehend exhibition information at different ages.

According to Margo Ngawa Neale (2022), the lead Indigenous curator at the National Museum, Aboriginal culture is primarily transmitted through performance, representing an embodied knowledge system. The *Connection* exhibition aligns closely with this notion, incorporating performative and cinematic elements. As immersive museum exhibitions gain prominence, the digital representation of Australian First Nations art presents a new challenge that prompts us to consider innovative design strategies. These strategies may involve virtual Aboriginal art and the incorporation of visitors' multisensory experiences within a multi-channel virtual space. The goal is to explore ways of effectively conveying the historical, cultural, and economic values of Aboriginal art to a broader community.

The arrangement of immersive elements within physical museum spaces, as well as virtual environments, requires innovative design practices. Recent discussions on designing immersive museum exhibits emphasize the creation of engaging experiences and the presentation of dynamic artworks (Lewi et al. 2020). While the digital representation of Australian Aboriginal painting art has achieved success, further research is needed to explore which immersive technologies can enhance greater connectivity, accessibility, and visibility (Geismar 2018) for other forms of Aboriginal art, including crafts. Additionally, there is a need to consider how the process of designing immersive exhibitions can be conducted within the cultural framework of Australian Aboriginal communities, ensuring the production of specific digital representations of Aboriginal art in the museum context.

## 4. Grande Experiences: Focusing on Visitors' Immersive Engagement

### 4.1. Grande Experiences' Related Works

*Connection: Songlines from Australia's First Peoples* represents the 250th immersive experience brought to life by Grande Experiences. As the owners and operators of THE LUME Melbourne and a renowned touring company behind successful exhibitions such as Van Gogh Alive, Monet and Friends, and Da Vinci Alive, the Grande Experiences company has been captivating audiences worldwide since its establishment in 2006. With a remarkable track record, they have provided unforgettable experiences to over 23 million visitors across the globe. In 2020, the company underwent a name change, transitioning from Grande Exhibitions to Grande Experiences. This new name reflects their evolving focus, which extends beyond enhancing the presentation of art, culture, and science, to fully integrating innovative narratives, digital technology, interactive elements, and immersive sensory experiences. This change also underscores the company's renewed guiding principles, which prioritize audience-centric on-site experiences while bridging the realms of art and entertainment.

Pan (2021) categorizes three types of curatorial approaches that the Grande Experiences company has conducted in few different exhibition activities (see Table 1).

The Grande Experiences company's high-tech display systems can be installed efficiently to adapt to various spatial conditions, making their digitized immersive environment distinctive and highly competitive in the commercial realm (Pan 2021). Technology-driven multisensory design has emerged as an innovative approach to delivering narrative content through immersive experiences (Dal Falco and Vassos 2017). According to Collin-Lachaud and Passebois (2008), immersive technologies play a significant role in visitor participation in museum activities. By allowing visitors to choose their own path through an exhibition, immersive technologies enable greater independence and the creation of personalized learning experiences. Within immersive museum exhibitions, visitors engage in a captivating and multisensory experience that stimulates various channels of the human brain (Pascual-Leone and Hamilton 2001). Sensory engagement in museums typically involves touch, sound, smell, and taste as visitors interact with the collections. Dudley (2011) expanded our understanding of museum artifacts by emphasizing the importance of aesthetics in shaping the sensory experiences and cognitive engagement of visitors. Pallasmaa (2014) demonstrated that exhibition design can influence visitors' perceptions and emotions towards museum collections. Immersive exhibitions leverage digital technology to deliver synthetic immersion and novel experiences to visitors in virtual spaces, using multiple devices to interpret signals and recreate the original exhibits (Kim et al. 2022). Through technology, exhibition elements are interconnected in a cyclical structure, providing expanded and intensified sensory experiences for visitors (Lee et al. 2021). In immersive exhibitions, visitors become integrated with the artwork, experiencing a given worldview through real imitations and metaphorical image formation within a liberated environment (Lee et al. 2019).

**Table 1.** Three types of Grande Experiences' curatorial approaches and examples.

| Grande Experiences' Curatorial Approaches | Type 1 | Type 2 | Type 3 |
|---|---|---|---|
| Descriptions | The display utilizes artifact-based interactive elements complemented by immersive multimedia presentations. | On-site interactive activities enhanced by immersive multimedia displays. | The display is entirely multimedia-based, designed to stimulate heightened multisensory and immersive experiences. |
| Examples | Leonardo da Vinci Collection (Italy, 2020) | Alice's Adventures in Wonderland (Global, 2016–) | Van Gogh Alive (Global, 2014–) |
| Details | Using SENSORY4 technology, the system integrates 360° large screens, high-definition projectors, and Dolby surround sound to showcase multi-channel moving images and video footage. It is accompanied by cinema-quality audio narration and music, creating a digitally immersive and multisensory space. Inviting audiences to touch and interact with the restored models. | The entire exhibition encompasses 37 projectors, 29 separate screens, 54 m of blended projection, and 3 distinct PA systems. All these elements operate in full synchronization, running on a 60-min automated loop. The experience immerses visitors in a super large format multimedia environment, taking them on a journey alongside Alice as she embarks on her adventures in Wonderland. It offers an immersive storytelling experience. | The immersive experience relies on advanced digital technologies, including multichannel display screens, all-surface projection, and digitally controlled surround sounds. These elements work together to create an immersive environment that stimulates a collective sense of space, presence, movement, sight, hearing, smell, and touch. |

Immersive technologies place a strong emphasis on the body, incorporating body motion into the functionality of devices (Grammatikopoulou 2017). By observing how visitors interact with artworks in immersive environments, we can observe the adoption of new roles that are more playful, active, and open to communication within museum settings (Grammatikopoulou 2017). Falk et al. (2004) use the term "whole body" to describe immersive replicas of objects or phenomena that are larger than life, allowing visitors to physically enter and engage with the replicas, resulting in a multisensory and kinesthetic experience (Dancstep et al. 2015). Immersive art exhibitions, by encouraging visitors to engage in an experiential process of physical exploration and discovery, can be seen as an extension of the real-world experience of observing artworks in physical spaces. By presenting artworks in an immersive space that requires physical movement or poses a challenge to visitors' physical abilities, the digital experience embodies visitors' core cognition in perceiving the value of artworks. Visitors have diverse needs, and even the same individual can experience the same exhibition multiple times, each with a different approach and varying expectations. The most satisfying exhibitions for visitors are those that resonate with their experiences and provide information in ways that validate and enhance their worldview (Doering and Pekarik 1996). According to Kelly (2003), visitors enter exhibitions with their own agenda and perspectives, particularly if the subject matter is current and significant. Some individuals seek affirmation of their existing views, while others are more open to exploring different perspectives and reconsidering their opinions. However, it is essential to acknowledge that visitors maintain control over their viewpoints and the willingness to change, rather than placing that responsibility on the museum.

One of the intriguing discussions in immersive virtual museum exhibitions revolves around striking a balance between visitor entertainment and educational experiences. Museums must effectively utilize their collections to both educate and entertain their audiences, recognizing that entertainment can be a means to facilitate education (Beeho and Prentice 1995). Much of the literature exploring education and entertainment within

the museum context emphasizes the differing perspectives of adult museum visitors regarding learning, education, and entertainment (Kelly 2003; Roppola 2012). Visitors possess the autonomy to freely choose their learning experiences, selecting and controlling content and learning methods based on their interests and needs (Falk and Dierking 2013). Hall and Bannon (2006) suggest that museum-based exhibitions are characterized by various factors, such as materiality, narration, sociality, activity, and multimodality. Studies have demonstrated that these factors enable visitors to engage with the exhibits while appreciating the provided context and knowledge actively and meaningfully (Perry 2012).

Indeed, visitors' preferences vary when it comes to exploring exhibits from diverse cultural backgrounds. Some individuals enjoy spending extended periods of time delving into specific exhibits, while others prefer a more concise tour, sampling a little bit of everything. It is crucial to consider how to encourage active participation in immersive exhibitions and how to engage visitors on a deeper level than ever experienced before within physical spaces. The philosophy guiding the design of immersive exhibits revolves around understanding visitors' needs and expectations. It is also essential to recognize the evolution of visitors' expectations, transitioning from a static viewing experience to immersive interactions within the context of a museum.

The Grande Experiences company's projects explore the conceptual intersections of art, nature, technology, and human experiences. They offer a prototype that utilizes immersive technologies to push the boundaries of artistic presentations. In turn, this prototype showcases the potential of integrating technology with digitized art, expanding the possibilities for immersive and innovative displays. The Grande Experiences company's projects not only showcase the close connection between artworks and humans but also demonstrate that reciprocal sensory interactions can be achieved without relying solely on immersive technologies. Through unique curatorial designs, the Grande Experiences company explores ways to create immersive experiences where visitors can engage with artworks and their surroundings, fostering a deeper connection and interaction that goes beyond the digital realm. Generally, the Grande Experiences company pioneers a new frontier in virtual museum exhibitions, pushing the boundaries of immersive and multisensory experiences to unprecedented levels. From the logical and conceptual aspects of the creation process to the exhibition of its final form, the Grande Experiences company introduces innovative strategies and models that have served as inspiration for key arguments presented in this study. The groundbreaking work of Grande Experiences redefines the possibilities of art, opening up new spaces for exploration and challenging traditional notions of artistic expression.

*4.2. Connections between Grande Experiences and the Immersive Exhibition of Australian Aboriginal Art*

Grande Experiences is one of a few Australian companies specializing in immersive exhibitions, with a proven track record of producing immersive experiences in collaboration with various cultural institutions around the world, including the National Museum of Australia, Kensington Gardens in UK, and Dewey Centre in China.

4.2.1. Contributions of Grande Experiences

Grande Experiences' selection was attributed to their extensive expertise and experience in creating immersive exhibitions of Australian Aboriginal art. Their prior projects and partnerships have showcased their capacity to handle intricate and culturally sensitive materials, establishing them as a reliable partner. Additionally, Grande Experiences' advanced technological capabilities and innovative solutions align well with the National Museum of Australia's vision. Their proficiency in immersive technologies allowed them to provide visitors with a cutting-edge and immersive experience. Furthermore, Grande Experiences have demonstrated a collaborative and consultative approach in their work with Aboriginal communities, artists, and cultural experts in *Connection: Songlines from Australia's First*

*Peoples*. This approach ensured the direct involvement of Aboriginal communities in the creation process, helping to prevent any misinterpretation of cultural materials.

### 4.2.2. Risks Aversion

There were risks associated with the immersive exhibition that Grande Experiences might inadvertently sensationalize or misinterpret cultural material. To mitigate the risks, it was imperative to conduct thorough research and engage in collaboration with Aboriginal communities. This ensured that the representations were both respectful and accurate. Moreover, questions pertaining to the ownership and utilization of cultural materials, as well as the fair compensation of Aboriginal artists and communities, should be addressed transparently (Yerkovich 2016).

Furthermore, it is of utmost importance to actively seek critical reviews of the exhibition, especially from Aboriginal communities and cultural experts. This provides valuable insights into whether the immersive experience aligns with their cultural values and expectations. Beyond the launch of the exhibition, the collaboration between the National Museum of Australia and Grande Experiences should be ongoing. This includes the establishment of continuous dialogues and feedback mechanisms to address concerns, implement necessary adjustments, and uphold the long-term integrity of the exhibition (Museums Australia 2005).

While Grande Experiences has a commendable track record, the selection of any company to produce immersive exhibitions of Aboriginal art should be based on a combination of factors, including expertise, technological capabilities, collaborative approaches, and a commitment to cultural sensitivity. It is of paramount importance to consider the potential impact and implications of these exhibitions and engage in ongoing dialogues and critical discussions to ensure the respectful representation of Aboriginal cultures. Critical reviews and feedback from Aboriginal communities and cultural experts play a significant role in this process.

## 5. Connecting Humans and Aboriginal Art in Digital Realm

Virtual museum exhibitions today have evolved into sophisticated digital renditions of real-life events or objects. Leveraging the capabilities of multimedia and immersive technologies, these digital representations create simulated environments accessible through websites and aim to provide the public with a comparable level of satisfaction as physical exhibitions. Through the integration of immersive technologies such as 360-degree tours and augmented, virtual, mixed, and extended reality, the public is able to explore and engage with objects or entire exhibition spaces from multiple dimensions, regardless of the user's physical location. In this immersive landscape, virtual exhibitions have the potential to expand the visibility of and interactions with virtual exhibits, thus transcending the confines of physical space.

In physical exhibitions of contemporary Aboriginal art, collaborative curatorial approaches are employed to amplify the significance of the featured objects, in turn facilitating the promotion and preservation of Aboriginal art knowledge. Aboriginal artists are increasingly embracing visual and digital mediums, such as photography, video, blogging, and social media, as platforms to showcase their work, exchange creative and technical ideas, foster communities, and generate business opportunities (Black 2005). By combining a curatorial framework with Aboriginal artists' increased interest in digital technologies, immersive virtual exhibitions centered on Aboriginal art are valuable means to communicate knowledge about and the heritage of the field to the public. In such simulated exhibition environments, immersive technologies play a pivotal role in the digital representation of Aboriginal art, transforming the objects into digital collections within an intricately interconnected and multidimensional space. These virtual settings intertwine diverse information about the pieces and provide the public with an opportunity to engage with a range of virtual Aboriginal art, which can be explored from various angles and perspectives.

A fundamental question emerges concerning the most effective approach to the digital representation of Aboriginal art in immersive virtual exhibition environments. The recommended method, as explored in this study, involves the creation of a three-dimensional model for each object, which, when combined with immersive technologies, allows the public to engage with the object through multiple senses, thereby transcending the scope of mere visual appreciation. In doing so, this investigation underscores the importance of immersive technologies, not only for digitally capturing the geometry and measurements of Aboriginal art, but also for effectively conveying its perceptual qualities and cultural significance.

The emergence and advancement of immersive technologies have revolutionized the possibilities for efficient information exchange and audience interaction in virtual exhibitions. Building upon the analysis presented above, the digital representation of Aboriginal art can be approached from three distinct dimensions: The first pertains to practicality, emphasizing the materials and functional aspects of Aboriginal art. The second explores cognition, encompassing Aboriginal art knowledge and the complex processes of cognitive and analytical thinking associated with it. The third delves into ontology, encapsulating the authenticity of Aboriginal art and its social and cultural significance within human life.

From a practical perspective, immersive virtual exhibitions centered around Aboriginal art focus on digitally representing tangible qualities of artworks, such as the materials and textures, utilizing appropriate technologies. For example, virtual reality (VR) has the potential to grant audiences access to objects that may otherwise be inaccessible (tom Dieck and Jung 2019), while simultaneously offering reflective learning experiences within the realm of cultural heritage (Han et al. 2018). This technology also enables personalized and tailored cultural heritage experiences, providing a unique and customized digital experience for the audience (tom Dieck and Jung 2019). By leveraging the power of immersive technologies, diverse cultural resources and a wealth of Aboriginal art-related information can be seamlessly integrated, enriching the audience's digital experience within a virtual exhibition space.

In terms of cognition, immersive technologies have the capacity to digitally document and contextualize the knowledge, skills, and cognitive systems inherent in Aboriginal art by connecting these intangible elements to virtual objects. The aim of digital representation is to transform these elements into explicit information, enabling immersive storytelling that enhances the audience's digital experience. By designing a digital narrative that encompasses the entirety of Aboriginal art, the public is able to explore and navigate the interconnected realms of this heritage.

From an ontological standpoint, the digital representation of Aboriginal art revolves around the creation of an authentic experience, fostering interactive engagement between the public and the exhibited objects. Immersive technologies facilitate the seamless integration of tangible and intangible elements of Aboriginal art, promoting a profound understanding of its material aspects and cultivating an awareness of its historical and cultural authenticity. To effectively convey the historical and cultural values of Aboriginal art to the broader public, it is essential to explore new approaches to digital representation that incorporate virtual Aboriginal artworks and engage the public's multisensory experience within an immersive virtual exhibition space. Additionally, by acknowledging the social, cultural, and commercial values associated with Aboriginal art, the digital representation of Aboriginal art can emphasize sustainability and the extended production timelines involved (Cantista and Delille 2023), encouraging the public to appreciate the artistic and cultural values intrinsic to Aboriginal art.

One of the distinctive attributes of Aboriginal art lies in its ability to continually redefine the dynamic interplay between Aboriginal artists, Aboriginal artworks, and consumers. By transforming commercial value into authentic value, diverse Aboriginal artworks serve as conduits for transmitting their stories (Cantista and Delille 2023). Immersive technologies play a vital role in enabling the public to engage themselves with the rich history and

culture associated with Aboriginal art. When attempting to represent the intangible aspects of Aboriginal art, designers of immersive virtual exhibitions are encouraged to explore new approaches that encompass idea development, conceptualization, and multisensory design. An immersive virtual exhibition dedicated to Aboriginal art should seek to enhance multisensory access and digital representation of its diverse facets. In general, immersive virtual exhibitions provide unique and promising opportunities to reconstruct and visualize embodied knowledge using digital materials drawn from the realm of Aboriginal art, underscoring the need for further research in this compelling field.

## 6. Conclusions

An increasing number of Australian museums are dedicated to providing visitors with an unforgettable experience by designing immersive scenarios that immerse them in the captivating stories of Australian Aboriginal art. Digital representation of Aboriginal art emerges as an approach that engages visitors on an emotional and cognitive level, effectively stimulating their senses within the museum's collections. Museum visitors continue to have high expectations of gaining insight into Aboriginal history, culture, and art through the utilization of advanced technology. To meet these expectations, museum curators and exhibition designers are encouraged to embrace new methodologies—from idea development and conceptual creation to storytelling and sensory design—in order to effectively represent the spiritual world of Australian Aboriginal artists. By promoting multisensory access to the rich diversity of Australian Aboriginal art, immersive exhibitions offer a unique opportunity to reconstruct and visualize embodied knowledge using digital materials. This potential requires further exploration in the field.

In conclusion, this study explores the role of immersive technologies in digitally representing Australian Aboriginal art within the museum context, considering its significant historical and cultural value. The focus is on design as an innovative approach that imbues Australian Aboriginal art with meaning in immersive museum exhibitions. However, it is important to acknowledge the existing limitations, particularly the gaps that exist between digital representation, immersive interactions, and the depth of rational design thinking. These gaps highlight the need for further exploration in terms of creating digital experiences that cater to diverse social groups, effectively representing the value of Australian Aboriginal art in immersive museum exhibitions while respecting the multicultural backgrounds of visitors.

**Author Contributions:** Methodology, R.Z.; Investigation, R.Z.; Resources, R.Z. and F.P.; Writing—original draft, R.Z.; Writing—review & editing, F.P.; Supervision, F.P. All authors have read and agreed to the published version of the manuscript.

**Funding:** This research received no external funding.

**Data Availability Statement:** No new data were created or analyzed in this study. Data sharing is not applicable to this article.

**Conflicts of Interest:** The authors declare no conflict of interest.

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
