# Peer review of "Connection: Digitally Representing Australian Aboriginal Art through the Immersive Virtual Museum Exhibition"

_arts, 1993_

Round 1

Reviewer 1 Report

Comments and Suggestions for Authors

This manuscript presents an exhibition that employs large-scale projections, cutting-edge light and sound technology to offer a mesmerizing glimpse into the intricate network of Australian Aboriginal art. This manuscript examines the Connection exhibition as a digital storytelling platform by exploring the a company’s approach to the digital replication of Australian Aboriginal art, with a focus on the connection between humans and nature in different exhibition spaces.

Unfortunately, the terms of "immersion", "virtual exhibition", "virtual space" are being used abusively in a wrong way. These terms are related with immersive technologies, such as extended reality (virtual reality, mixed reality, augmented reality). These technologies are not presented on this manuscript.

Furthermore, the manuscript is presented like a promotional or advertising announcement of a company, rather than a scientific article. There are no discrete and clear steps in methodology that this exhibition could be replicated. In addition, there are no evidences and results that are based on the experience of the real visitors, such as questionnaires and surveys. In conclusion, I do not feel confident to accept the manuscript for publication, as it does not offer any significant contribution in the scientific area.

Author Response

Thank you very much for taking the time to review this manuscript. 

The purpose of this paper is to construct a framework that elucidates how Australian Aboriginal Art can be effectively digitally represented using immersive technologies in museum context. This is achieved by:

1) Assessing the Connection exhibition from a design perspective (Section 2).

2) Acknowledging the contributions of Grande Experiences (Section 3).

3) Offering a comprehensive checklist for the design and implementation of future immersive museum exhibitions (Section 4).

For highlighting the importance of Australian Aboriginal art and the Connection Exhibition in the field, we have added a new section of "Understanding Australian Aboriginal Art". 

Since this paper primarily centers around the analysis of the Connection exhibition and the immersive projection technology employed within it, we do not delve into discussions about other immersive technologies. Nevertheless, we have introduced a few examples of the application of other immersive technologies, such as VR, in Section 4.

Reviewer 2 Report

Comments and Suggestions for Authors

This paper presents the case study Connection: Songlines from Australia’s First Peoples, an immersive virtual exhibition of Australian Aboriginal art launched by the National Museum of Australia in 2022. Starting from it, this paper demonstrates that through a digital, multy-sensory and immersive storytelling it is possible to foster visitors’ engagement, inclusivity and interaction and promote, at the same time, a deep comprehension of the contents of the exhibition. After an introduction where the context and background in which this study is set is explained, the corpus of the text contains a detailed and well referenced description of the case study. The description is logical, schematic and enriched with a table and two images. In this regard, it is recommended to add some more pictures of the museum, in order to help the reader to have a better understanding of its uniqueness. The language is clear and the speech structure is coherent and cohesive. In the conclusions, the limitations of the case study are illustrated to open up to possible future developments.

Author Response

Thank you very much for taking the time to review the manuscript.

Reviewer 3 Report

Comments and Suggestions for Authors

This article presents a topic of relevance in the field of museum studies with specific reference to the use of technology to enhance visitor engagement. Of particular interest in the article is the concept of designing a multi-level visitor engagement experience. With reference to the proposed research, it is advisable, in the future, to deepen studies on museum didactics and in particular those focusing on the use of technology to foster meaningful learning experiences in relation to different audiences.

Author Response

Thank you very much for taking the time to review the manuscript. In future research, we will deepen studies on museum didactics and in particular those focusing on the use of technology to foster meaningful learning experiences in relation to different audiences.